# Roughness of Production Conditions: Does It Really Affect Stability of IgG-Based Antivenoms?

**DOI:** 10.3390/toxins14070483

**Published:** 2022-07-13

**Authors:** Sanja Mateljak Lukačević, Tihana Kurtović, Juraj Borić, Beata Halassy

**Affiliations:** 1Centre for Research and Knowledge Transfer in Biotechnology, University of Zagreb, Rockefellerova 10, HR-10000 Zagreb, Croatia; tkurtovi@unizg.hr (T.K.); jboric@unizg.hr (J.B.); 2Center of Excellence for Virus Immunology and Vaccines, CERVirVac, Rockefellerova 10, HR-10000 Zagreb, Croatia

**Keywords:** plasma processing, IgG antivenom, aggregates, thermal stability

## Abstract

Antivenoms contain either pure animal IgGs or their fragments as an active substance, and are the only specific therapeutics against envenomation arising from snakebites. Although they are highly needed, the low sustainability of such preparations’ manufacture causes constant global shortages. One reason for this is the stability of the product, which contributes not only to the manufacture sustainability, but the product safety as well. It has been hypothesized that the roughness of conditions to which IgGs are exposed during downstream purification disturbs their conformation, making them prone to aggregation, particularly after exposure to secondary stress. The aim of this research was to investigate how the roughness of the downstream purification conditions influences the stability properties of purified IgGs. For this purpose, equine IgGs were extracted from unique hyperimmune plasma by two mild condition-based operational procedures (anion-exchange chromatography and caprylic acid precipitation) and three rougher ones (ammonium sulphate precipitation, cation-exchange chromatography and protein A affinity chromatography). The stability of the refined preparations was studied under non-optimal storage conditions (37 °C, 42 °C, and a transiently lower pH) by monitoring changes in the aggregate content and thermal stability of the pure IgGs. Mild purification protocols generated IgG samples with a lower aggregate share in comparison to the rougher ones. Their tendency for further aggregation was significantly associated with the initial aggregate share. The thermal stability of IgG molecules and the aggregate content in refined samples were inversely correlated. Since the initial proportion of aggregates in the samples was influenced by the operating conditions, we have shown a strong indication that each of them also indirectly affected the stability of the final preparations. This suggests that mild condition-based refinement protocols indeed generate more stable IgGs.

## 1. Introduction

Snakebite envenoming constitutes a significant public health burden that mostly affects the poor rural communities of tropical regions. As such, it causes high morbidity and mortality, mainly due to limited access to adequate health services and the insufficiency of safe and effective antivenoms [1,2,3], which are the only specific treatment tool for this medical issue [4]. However, the manufacture of antivenoms containing either whole animal-derived pure IgGs or their fragments is of low sustainability due to various reasons [5]. One of them is the stability of their active components, a feature that significantly affects the quality as well as the safety of these therapeutics [6]. 

Traditionally, whole IgGs are purified from animal hyperimmune plasma by diverse refinement methods based on different biochemical principles, whose operational conditions might have variously pronounced effects on IgG stability. It has been hypothesized that procedures which employ more aggressive operational conditions, such as the precipitation of IgGs with a high salt concentration or their binding to chromatographic carriers followed by elution with buffers of increased ionic strength or low pH, lead to permanent or transient conformational disturbances with an impact on the stability of IgGs and, consequently, their tendency for aggregation [7]. This becomes particularly evident if they are subsequently exposed to secondary stress [8]. On the other hand, by applying mild condition-based purification procedures, the structural integrity of IgGs remains intact, presumably due to the fact that they are kept in the solution throughout the fractionation process [9,10,11]. 

Our previous study has demonstrated that available and widely used refinement concepts have different influences on certain quality-related features of the final product. Stability-wise, the results showed that preparations obtained by operational procedures which employed rough conditions (cation-exchange chromatography (CEX), protein A affinity chromatography (AC) and ammonium sulphate precipitation (ASP)) contained increased aggregate content in comparison to preparations refined by gentle protocols (caprylic acid precipitation (CAP) and anion-exchange chromatography (AEX)) [12]. This indicates that the rough protocols had a negative impact on the active component structure, and consequently, its stability [13]. 

To extend this research, we aimed to investigate the influence of the aforementioned biochemical principles on the stability properties of pure IgGs. We monitored the changes in monomer content and aggregate share and the shift in the melting temperatures of the pure IgGs in the preparations after exposing them to various adverse conditions (storage at the suboptimal temperatures of 37 °C and 42 °C, and the transient lowering of pH) as additional stress factors that might elicit possible differences in their stability. The quantity of aggregates that formed in the samples during IgG isolation proved to be a major driving factor for further aggregation under adverse storage conditions. Since the operational conditions of the refinement procedures directly affected the proportion of aggregates in the final products, it can be concluded that each of them also indirectly affected the stability of the pure IgGs. Therefore, when designing the process of IgG extraction from hyperimmune equine plasma, choosing steps that reduce the aggregate formation to the lowest degree is of the utmost importance.

## 2. Results

We investigated the stability of the active therapeutic component in samples prepared from the unique pool of *Vipera ammodytes ammodytes* (*Vaa*)-specific hyperimmune horse plasma (HHP) by two mild (CAP and AEX) and three rougher (CEX, ASP and AC) purification protocols. The refined preparations, formulated in the same buffer, were simultaneously monitored for changes in the monomer content and aggregate share, as well as shifts in the melting temperatures (*T_m_*) of IgGs after exposure to various unfavorable storage conditions. Our previously published results showed that CAP and AC protocols generate highly pure products (>90%) in a single step, while the preparations obtained by AEX, CEX and ASP procedures contain high numbers of impurities (17–27%), making them unsuitable for stability analyses by thermal shift assay. Therefore, those less pure samples (AEX1, CEX1 and ASP1) were additionally refined using a caprylic acid precipitation step (AEX2, CEX2 and ASP2), the implementation of which sufficiently improved their purity (>97%) [12].

### 2.1. One-Month Storage at 37 °C

Among the used purification procedures, the AC protocol stood out as the one generating the product with the lowest content of IgG monomers (89%) and the highest share of aggregates (8.8%) (Table 1, before storage). Its *T_m_* value was lower by 1.5 °C in comparison to those measured for other samples (Figure 1). The proportion of IgG monomers and aggregates (Table 1), as well as their *T_m_* values (Figure 1), did not change substantially after one month of storage at 37 °C.

Such findings indicated that the IgG samples needed to be placed in rougher environmental conditions in order to provoke the disclosure of possible structural discrepancies that occur during downstream processing and consequently might lead to stability differences. 

### 2.2. Three-Month Storage at Different Temperatures

In further research, the storage time was extended from one to three months at three different temperatures (4, 37 and 42 °C). The lowest temperature (4 °C) had no effect on the IgG content and aggregate share, the values of which remained mostly unchanged regardless of the employed refinement protocol (Figure 2). However, storage at both the elevated temperatures (37 and 42 °C) resulted in a decrease in IgG monomers (Figure 2A) and an increase in aggregates (Figure 2B) for each type of IgG sample. The amount of aggregates (Figure 3A) and their level of increase (Figure 3B) in analyzed preparations during storage at 37 and 42 °C were positively correlated to their initial share. This phenomenon was even more pronounced in the preparations kept at 42 °C, among which the sample from AC processing had the highest IgG monomer loss (9.4%) (Figure 2A) and the largest increase in aggregate formation (6.7%) (Figure 2B). 

The thermal stability of IgGs in all the samples stored at 4 °C remained almost unchanged, as their melting temperatures decreased by less than 1 °C. Opposite to that, storage at 37 and 42 °C negatively affected the thermal stability of the samples and caused a decrease in the *T_m_* values. A particularly prominent reduction in the melting temperature of 2.7 °C was observed for IgGs in the sample from ASP processing that was stored at 42 °C (Figure 4).

A negative correlation tendency between the aggregate content and the thermal stability of IgGs from the respective preparations, measured prior to and after three months of storage at three different temperatures, was evident (Figure 5).

### 2.3. The Influence of Transient Exposure to Low pH on the Stability Parameter

One set of pure IgG preparations was transiently exposed (for 2 h) to acidic conditions (pH 2.0), as another stress factor, and subsequently stored at 37 °C. The extremely low pH resulted in a slight decrease in IgG monomer content in most of the analyzed samples (Figure 6A) and an increase in aggregate formations in all of them (Figure 6B). This trend became even more pronounced after three months of storage at an elevated temperature (Figure 6). Preparations that contained less aggregates before the exposure to low pH contained fewer aggregates after the acidification and storage at higher temperature (37 °C), and vice versa (Figure 6B). 

The transient acidification unexpectedly induced a positive effect on the thermal stability of IgGs by increasing the melting temperatures in the majority of preparations. However, after being stored for three months at an elevated temperature (37 °C), a TSA analysis revealed a notable shift in the melting temperatures of IgGs towards lower values (1–3 °C) in all samples (Figure 7), in proportion to their initially formed aggregate contents (Figure 6B). 

By this point in the research, we had determined and repeatedly shown that the presence of aggregates in the samples formed during the refinement processes is the dominant factor that induces further aggregation of IgGs in proportion to their initial share when stored in unfavorable conditions. Still, we could not associate the tendency of IgGs to aggregate with the rougher or milder conditions that they were subjected to during the process of isolation from HHP. Therefore, in the last step of our investigation, we aimed to remove aggregates from the refined samples as much as possible in order to attribute possible differences between them, in terms of their stability and tendency for aggregation, exclusively to the disturbances in IgG monomer structure occurring as a consequence of the operational conditions of the purification process.

### 2.4. Stability of IgG Samples after the Reduction of Initial Aggregate Content

IgG samples, prepared at least nine times independently by caprylic acid precipitation (CAP) and protein A affinity chromatography (AC), were additionally purified by the AEX procedure. All preparations were analyzed for aggregate appearance, IgG monomer content and thermal stability before and after the three months of storage at the favorable 4 °C and the extremely unfavorable 42 °C. Simultaneously, the same stability parameters of one IgG sample set, exposed to the acidic environment and stored afterwards at 4 °C, were also examined. AC protocol-based products contained twice as many aggregates (6.9% ± 0.32%) as those from CAP processing (3.2% ± 0.04%) (Figure 8A), which was in accordance with previous experimental results. The additional purification by AEX significantly reduced the aggregates in both types of samples, by four-fold in the CAP AEX (0.7% ± 0.05%) and two-fold in the AC AEX (3.7% ± 0.33%). However, complete aggregate removal was not achieved (Figure 8A).

At the same time, the IgG monomer content increased (3–7%) in both types of highly purified preparations. Transient (for 2 h) low pH treatment (pH 2.0) caused a very large increase in aggregates (62% ± 0.22% in the CAP AEX pH samples and 66% ± 0.85% in the AC AEX pH samples) and also caused a significant decrease in IgG monomer content (34% ± 0.30% in the CAP AEX pH samples and 31% ± 1.23% in the AC AEX pH samples) (Figure 8A).

The three months of storage at 4 °C did not affect the proportion of IgG monomers and aggregates in the CAP AEX and AC AEX samples, while the higher temperature (42 °C) resulted in approximately a 4% increase in aggregates and a slight decrease in IgG monomers (>7%) for both types of preparations. Also, CAP AEX samples contained significantly lower numbers of aggregates compared to AC AEX after being stored at an elevated temperature (42 °C) (Figure 8B). The transient pH-lowering induced a very large increase in aggregates (≈60%), measured at the beginning of the stability testing (Figure 8A), the amount of which additionally increased by around 4% after the three months of storage at 4 °C (CAP AEX pH 4 °C and AC AEX pH 4 °C, Figure 8B). Thus, the total increases in the aggregates of the CAP AEX pH 4 °C and AC AEX pH 4 °C preparations were 65.1% and 64.3%, respectively, compared to the non-acidified ones.

The TSA analysis confirmed once again that the CAP protocol generates more thermally stable IgGs, the melting temperatures of which were 0.7% higher compared to the IgG molecules isolated by the AC method (Figure 9A). These results were in accordance with the quantity of aggregates present in the analyzed samples (Figure 8A). A further reduction in aggregates by the AEX step caused a proportional increase in the thermal stability of the IgGs in both types of samples by 0.6 °C (CAP AEX and AC AEX) in comparison to those not subjected to aggregate removal (CAP and AC) (Figure 9A). Although a very large quantity of aggregates was present in samples exposed to low pH (CAP AEX pH and AC AEX pH), their melting temperatures remained high and unchanged (Figure 9A).

Storage at 4 °C did not have any influence on the thermal stability of CAP and AC samples, or on that of the respective products obtained by an additional chromatographic step (CAP AEX, AC AEX) (Figure 9B). Namely, their *T_m_* values were equal to those measured at the beginning of the stability testing (Figure 9A). On the other hand, storage at 42 °C resulted in a reduced stability of highly purified IgGs in both types of samples, decreasing their melting temperatures by more than 2 °C compared to those stored at 4 °C. Acidification, as another unfavorable factor, contributed the most to the destabilization of immunoglobulins, reducing the *T_m_* values of AC AEX pH samples by 3.4 °C, and those of CAP AEX pH preparations by as much as 4.9 °C compared to the same samples unexposed to low pH (Figure 9B).

In the final experiment, the large number of analyzed samples enabled us to confirm a negative correlation between aggregate content and the thermal stability of IgGs (Figure 5 and Figure 10). The only exceptions were the acidified preparations, the *T_m_* values of which were equal to those of the untreated ones, which were determined immediately after the exposure to low pH (data not shown), despite the large quantity of aggregates (over 60%) they contained (Figure 8A). However, after three months, a negative correlation between these two parameters was observed in all IgG samples, including those that were transiently acidified. The high dependence of these two features proved to be linear only for preparations containing up to 10% of aggregates, both before and after the completion of the stability testing (Figure 10).

## 3. Discussion

The literature data imply that rougher isolation methods destabilize the structure of IgGs, making them less stable and more prone to aggregation [7,13,14,15,16], and that such an impact can be avoided by applying more gentle purification procedures. In our previous research, we have found that the operational conditions of two mild (CAP and AEX) and three more aggressive (CEX, ASP and AC) refinement processing strategies had different effects on the IgG subclass composition, the venom-specific protective efficacy, the profile of retained impurities and the thermal stability of immunoglobulins, as well as the share of aggregates [12], whose presence in the refined preparations is unwanted due to their ability to cause side effects in clinical settings [17,18,19]. Here, we report the findings of an extended investigation examining pure IgG preparations, obtained by the same aforementioned refinement protocols, after being exposed to unfavorable storage conditions as an additional stress factor. Our aim was to explore whether the downstream processing operational conditions influence the stability of IgGs, and consequently their tendency for aggregation. 

The preliminary experiments were performed in order to collect screening data on the differences in the stability of pure IgGs under the influence of various adverse environmental conditions (storage at elevated temperatures and low pH treatment). At the beginning, all five IgG samples isolated by different biochemical principles were kept for one month at 37 °C, since such an environment mimics a stability study over a 4-year-long period at 4 °C [20]. The proportions of IgG monomers and aggregates in the preparations remained mostly unaltered, irrespective of the applied refinement method (Table 1), and so did the melting temperatures of pure IgGs (Figure 1), indicating a remarkable stability of immunoglobulins as already reported [6,21]. Therefore, it was necessary to subject IgGs to rougher experimental conditions, under which the differences in their stability might be revealed in the form of progressive aggregation [22] as a consequence of their conformational alterations of various degrees. Besides 4 and 37 °C, samples were exposed to 42 °C, as an extremely unfavorable temperature, and the storage time was prolonged from one to three months, representing the shortest incubation period for stability testing that provides the preliminary information on a product’s quality, according to the regulatory guidelines [4].

As expected, the lowest temperature (4 °C) did not have a significant impact on the physicochemical properties of IgGs (Figure 2A,B and Figure 4), while both elevated temperatures (37 and 42 °C) negatively affected the quality of samples by inducing a shift in the melting temperatures towards lower values (Figure 4), a decrease in IgG monomer content (Figure 2A) and an increase in aggregates (Figure 2B). Similarly, Rojas et al. observed progressive aggregate formation in a liquid IgG-based antivenom, due to physicochemical changes in the active component that manifested in the form of temperature-dependent turbidity enhancement over a one-year timeframe [21]. In addition, we noticed that the long-term exposure of IgGs to adverse-temperature environments caused an increase in aggregates in proportion to the initial share of aggregates in most of the preparations (Figure 3). The detection of such a strong positive correlation between the initial and final amounts of aggregates (Figure 3) indicates that the propensity towards further aggregation depends on the aggregate share formed under the influence of the employed purification protocol. The formation of more aggregates in IgG samples stored at 42 °C in comparison to those kept at 37 °C proves that a higher temperature accelerates the aggregation at a greater rate (Figure 2B and Figure 3). Such findings support the explanation of the growth of aggregates through nucleation, a process during which new IgG monomer units are added to the initially formed aggregate nuclei [23,24], which is further promoted and accelerated by exposure to elevated temperatures. Namely, increased temperatures partially unfold proteins, disclosing the hydrophobic amino acids that are usually hidden within their core and that represent “hot spots“ for aggregation. After being exposed to the solvent, such sequences tend to form strong irreversible bonds, which are particularly common between the β-folded structures [25,26] that are highly represented in immunoglobulins [27]. 

Since we aimed to relate the differences in IgG stability solely to the process-related disturbances, we had to cancel the effect of initially present aggregates by removing them from the preparations. The samples obtained by CAP and AC, selected as the representatives of milder and rougher operational condition procedures, were further refined by anion-exchange chromatography (AEX) as a procedure previously reported to be effective for that purpose [9,28]. The pH value of the binding buffer (5.0) was sufficiently distant from the pI value of immunoglobulins, ensuring their unimpeded passage through the column while the remaining protein components, together with aggregates, bound to the stationary phase. Nevertheless, complete aggregate removal was not accomplished, although the quantity of aggregates formed during the extraction of IgGs by the primary purification procedures was significantly reduced in both types of samples (CAP AEX and AC AEX) (Figure 8A), leading to an improvement in the thermal stability compared to their more aggregated variants (CAP and AC) (Figure 9A). Such results, which correspond to those obtained by monitoring the influence of three months of storage at diverse temperatures on the IgGs purified by five refinement protocols (Figure 2A and Figure 4), confirmed that the thermal stability of IgGs and the aggregate content in the final preparations correlate in a negative manner (Figure 5 and Figure 10), as we have already reported [12]. Contrary to this, another study did not find any correlation between these two features, probably because of the fact that the IgGs were prepared in different matrices for the determination of their melting temperatures, while in this work all the samples were analyzed in the same matrix [29]. 

At this point, it is worthy of emphasis that both determined correlations—the positive one between the initial and final aggregate share, and the negative one between the thermal stability of IgGs and the amount of aggregates in the refined samples—may serve as valuable tools in IgG-based antivenom production for the preliminary prediction of stability, and therefore the safety, of these pharmaceuticals. 

In an attempt to elucidate the differences in the conformational disturbances of IgGs that occurred during isolation procedures, we also explored the influence of low pH as another stress factor, which has the most significant effect on antibody aggregation according to the literature [30]. The transient acidification (2 h) induced an immediate increase in the number of aggregates in all preparations (Figure 6B), including those additionally purified by AEX (Figure 8A), which further progressed during the subsequent three months of storage at 37 °C and 4 °C (Figure 6B and Figure 8B, respectively). Unexpectedly, this did not affect the thermal stability of the samples refined by the additional AEX step (Figure 9A), despite the obvious enhancement of aggregate content (Figure 8A). Moreover, acidification even demonstrated an immediate beneficial impact on the majority of the samples obtained by primary extraction procedures, as it notably increased their melting temperatures in relation to the untreated ones (Figure 7). As discussed by Gao et al., a possible explanation for such a phenomenon might be the generation of more stable complex protein structures, which occurs as a consequence of environmental alterations. Namely, changes in the pH value of a solution may induce the formation of new molecular interactions between proteins through hydrogen and van der Waals bond formation. Such structural rearrangements occur as a protein attempts to achieve the most thermodynamically stable conformation, which could consequently be responsible for its increased thermal stability [31]. However, this observed phenomenon was transient, since after three months of of storage at 37 °C, the *T_m_* values of the IgGs in all the samples significantly decreased (Figure 7) in accordance with the increased share of aggregates (Figure 6B). Equally, the thermal stability of CAP AEX and AC AEX preparations significantly dropped, despite the fact that they were stored at a lower temperature (4 °C) (Figure 9B).

Based on the presented results, IgGs refined by any of the applied biochemical principles proved to be stable drugs under optimal conditions. However, suboptimal storage conditions revealed the differences in their stability, indicating that IgGs prepared by mild condition-based purification protocols contained low initial levels of aggregates and, as such, were more robust and less prone to aggregation. Since the operating conditions of the purification procedures directly affected the proportion of aggregates in the refined products, it is likely that each of them also indirectly affected the stability of IgGs. Therefore, when designing the production process, it is of paramount importance to choose only those steps that minimize the occurrence of aggregation, thus avoiding the need for additional steps of aggregate removal. In this way, the stable active substance of antivenom, with a long shelf-life, can be produced in only several effective steps. This would increase the cost-effectiveness, and consequently the sustainability, of antivenom manufacture. 

## 4. Materials and Methods

### 4.1. Animals, Snake Venom, Plasma Pool and Chemicals

The pool of *V. ammodytes ammodytes* (*Vaa*)-specific hyperimmune horse plasma (HHP) was provided by the Institute of Immunology Inc., Zagreb, Croatia. Caprylic acid (≥98%) was obtained from Sigma-Aldrich, USA. Sorbitol and Tris base were obtained from Merck, Darmstadt, Germany. 3-(N-morpholino)propanesulphonic acid (MOPS) and 2-(N-morpholino)ethanesulphonic acid (MES) monohydrate were obtained from AppliChem, Darmstadt, Germany. Sodium acetate was obtained from Fluka, Switzerland. Sodium hydroxide was obtained from T.T.T., Sveta Nedjelja, Croatia. Other chemicals for buffers and solutions were obtained from Kemika, Zagreb, Croatia, unless stated otherwise.

### 4.2. IgG Purification Procedures

The starting material for each fractionation procedure was a unique sample of *Vaa*-specific HHP, thermally treated for 1 h at 56 °C. After centrifugation at 3200 × *g* for 45 min at room temperature (RT), the pellet was discarded and the supernatant was subjected to the respective purification procedure. Caprylic acid (CA) precipitation of non-IgG proteins, ammonium sulphate precipitation (ASP), anion-exchange chromatography (AEX), cation-exchange chromatography (CEX) and protein A affinity chromatography (AC) were performed as described by Mateljak Lukačević et al. [12]. Samples of an unsatisfactory purity level (˂90%) obtained by the AEX, CEX and ASP protocols (AEX1, CEX1 and ASP1) were further refined by the CAP procedure (AEX2, CEX2 and ASP2) using caprylic acid, the concentration of which was adjusted according to the amount of retained impurities [12]. 

### 4.3. Diafiltration

IgG preparations from all purification procedures were diafiltrated using a Vivacell device with a 100 kDa molecular weight cut-off (MWCO) polyethersulfone membrane (Sartorius, Germany). Samples from ASP and CEX fractionation were first desalted by diafiltration into a 50 mM MES buffer with a pH of 5.5. All the final preparations, including those that were obtained by an additional step of caprylic acid precipitation for purity level improvement, were diafiltrated into a 0.2 M phosphate buffer with a pH of 6.0, ensuring matrix uniformity prior to further analysis. Several repeated cycles of concentration and dilution were performed to achieve approximately 4 logs of buffer exchange.

### 4.4. Removal of Aggregates from Pure IgG Preparations

Samples from CAP and AC processing, each independently prepared at least nine times, were diafiltrated into a 20 mM sodium acetate buffer with a pH of 5.0, as described in Section 4.3. Aggregate removal was performed by anion-exchange chromatography (AEX) on a CIMmultus QA column (*V* = 0.34 mL; BIA Separations, Ajdovščina, Slovenia) using the same buffer as the mobile phase. The samples were loaded in a volume of 2 mL per run. The bound proteins were eluted with 1 M NaCl. The flow rate was 1 mL min^−1^. The flow-through fractions of reduced aggregate content (CAP AEX and AC AEX) were diafiltrated into a 200 mM phosphate buffer with a pH of 6.0.

### 4.5. Thermal Shift Assay

A thermal shift assay (TSA) was employed for measuring the impact of different refinement protocols on the thermal stability of IgGs. Prior to its performance, the remaining impurities from the final products of unsatisfactory purity level were additionally processed by caprylic acid precipitation, as described in [12]. The concentration of the precipitating agent was optimized depending on the contamination degree, ranging from 0.5 to 2% (*V*/*V*). 

The reaction mixtures were prepared in optical tubes (Applied Biosystems, Waltham, MA, USA) by mixing 25 μL of highly pure IgG samples (1 mg mL^−1^), previously diafiltrated into a 0.2 M phosphate buffer with a pH of 6.0, using 20 μL of the buffer and 5 μL of Sypro Orange dye solution (10× concentrate; Molecular Probes, Eugene, OR, USA). Negative controls without protein samples were included as well. The TSA analysis was performed using a qPCR instrument (7500 Real Time PCR, Applied Biosystems, Waltham, MA, USA). The reaction mixtures were incubated for 10 min at 5 °C, followed by an increase in temperature at a heating rate of 1 °C min^−1^ until reaching 94 °C. *T_m_* values were determined from the obtained curves by nonlinear regression using GraphPad Prism software (version 5.00 for Windows, GraphPad Software, San Diego, CA, USA). Each sample was analyzed at least five times.

### 4.6. IgG Monomer and Aggregate Content Profiling

Size-exclusion high-performance liquid chromatography (SEC-HPLC) was applied to monitor the IgG monomer and aggregate content of the purified preparations. It was performed on a TSKGel G3000SWXL column (7.8 × 300 mm; Tosoh Bioscience, Tokyo, Japan) in a 0.1 M phosphate-sulphate running buffer with a pH of 6.6, at a flow rate of 0.5 mL min^−1^ and at RT on a Shimadzu HPLC system (Shimadzu, Kyoto, Japan). The samples (1 mg mL^−1^), pre-treated by centrifugation for particulate removal, were loaded in a volume of 50 μL. The absorbance was monitored at 280 nm. The standard proteins used for molecular weight determination were thyroglobulin (*M*_r_ 669,000), γ-globulin (*M*_r_ 150,000), ovalbumin (*M*_r_ 43,000) and ribonuclease A (*M*_r_ 13,700). A correction factor corresponding to the deviation in the molecular mass of the analyzed IgG, determined from its nominal value on the calibration curve, was included in the calculation.

### 4.7. Stability Testing Conditions

The stability of highly pure IgGs extracted by CA, ASP, CEX, AEX and AC protocols and diafiltrated into 0.2 M phosphate buffer with a pH of 6.0 (1 mg mL^−1^) was preliminary investigated by comparing the results obtained with TSA and SEC before and after one month of storage at 37 °C. Then, the influence of two additional temperatures (4 and 42 °C) was examined as well. 

The effect of an acidic environment was tested as follows. First, the pH value of the samples was lowered to 2.0 using 6 M HCl. After 2 h at RT, the pH value was raised to the initial one (pH 6.0) with 6 M NaOH. The transient acidification was followed by storage at 37 °C. One set of concerning CAP- and AC-based preparations, the aggregate content of which was reduced by the AEX step as described in Section 4.4., was stored at 4 °C. Another was first subjected to the above-described transient acidification (2 h, pH 2.0, RT) prior to storage at 42 °C. All samples were filtered through a polyvinylidene difluoride (PVDF) membrane filter (TPP, Switzerland) with a pore size of 0.22 μm prior to storage under the described conditions for 3 months.

### 4.8. Data Analysis

The results of each analysis are expressed as the average (arithmetic mean) of *n* measurements ± standard error (SE). The number of measurements for each analysis (*n*) and the number of analyzed samples are given. The correlation between the set of different assays, expressed as the *r* value, was calculated using the software Statistica 13.5 (StatSoft, TIBCO Software Inc., Palo Alto, CA, USA) with the uncertainty of measurements expressed as a 95% confidence interval. A two-sided Mann–Whitney U rank test with *p* < 0.05 was used for individual comparisons of the two groups of data. 

## Figures and Tables

**Figure 1 toxins-14-00483-f001:**
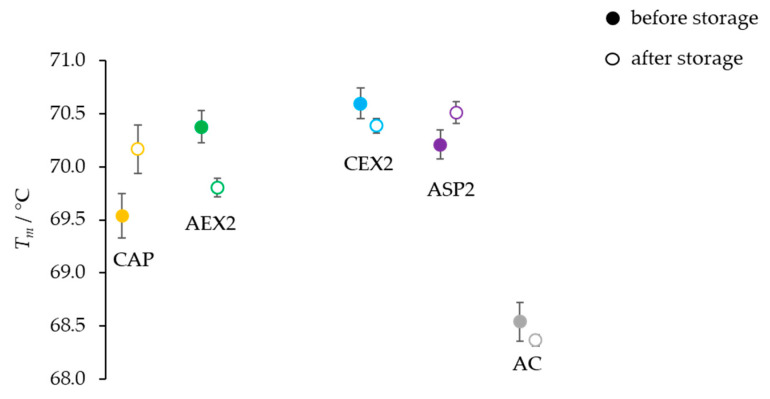
Thermal stability of highly pure IgG samples (*m* = 25 μg) obtained by CAP, AEX, CEX, ASP and AC protocols (one sample of each type) before and after one month of storage at 37 °C. Results are given as the mean of *T_m_* values ± SE (denoted by error bars) of at least 8 measurements.

**Figure 2 toxins-14-00483-f002:**
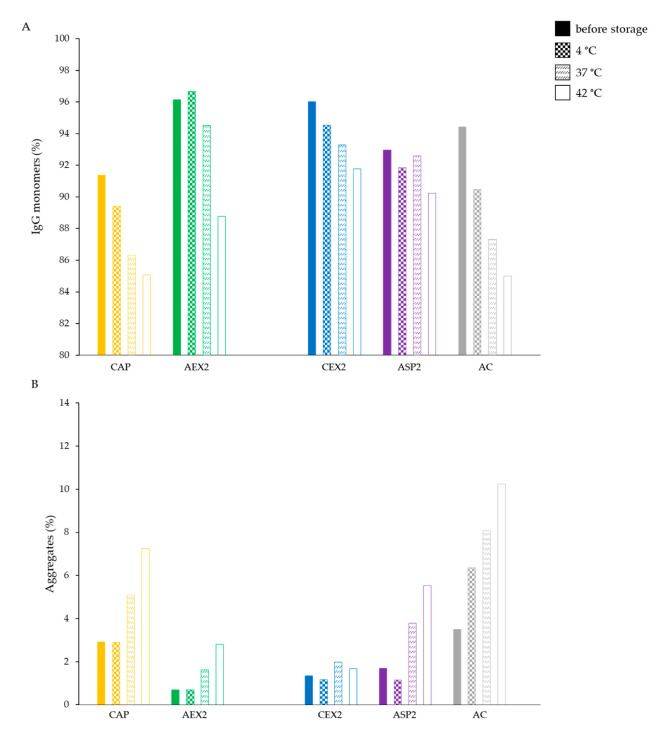
IgG monomers (**A**) and aggregate share (**B**) in IgG preparations (*c* = 1 mg mL^−1^) purified by CAP, AEX, CEX, ASP and AC refinement protocols before and after three months of storage at 4, 37 and 42 °C.

**Figure 3 toxins-14-00483-f003:**
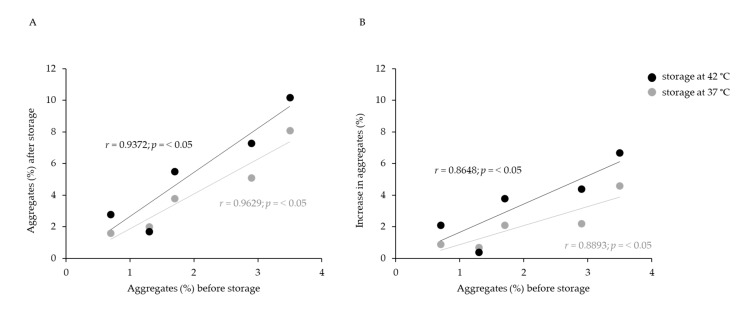
Initial aggregate content in IgG preparations representing a trigger for further aggregation demonstrated by positive corelation between their initial and final share in the samples stored at 37 °C and 42 °C during three months (**A**) as well as between aggregate increase and their initial share in the same preparations (**B**).

**Figure 4 toxins-14-00483-f004:**
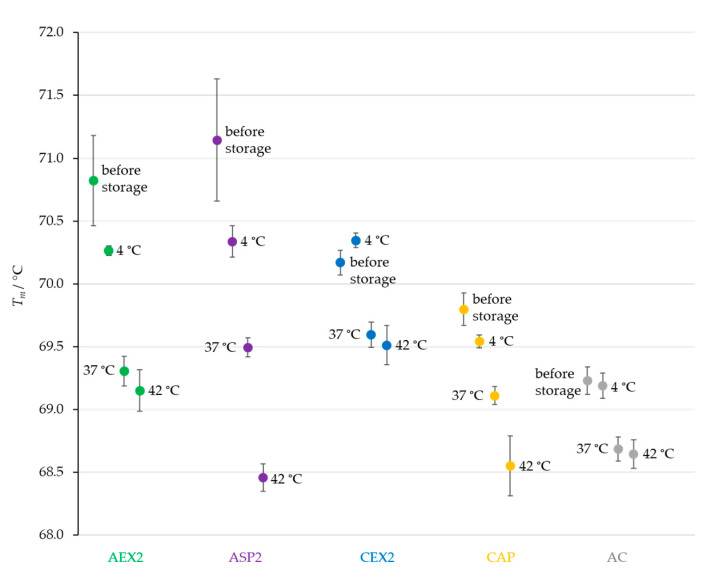
Changes in melting temperatures (*T_m_*) of pure IgG preparations (*m* = 25 μg) obtained by CAP, AEX, CEX, ASP and AC isolation methods (one sample of each type) before and after three months of storage at 4, 37 and 42 °C. Results are given as the mean of *T_m_* values ± SE (denoted by error bars) of at least 6 measurements.

**Figure 5 toxins-14-00483-f005:**
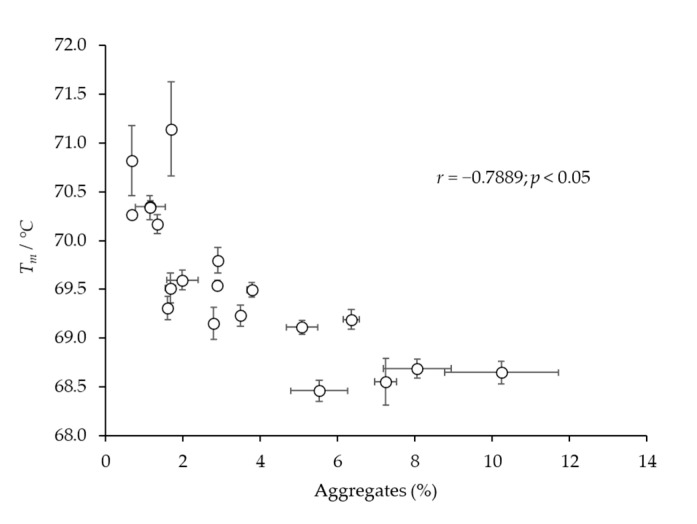
Negative correlation between *T_m_* values of IgG monomers and aggregate content in the purified preparations obtained by CAP, AEX, CEX, ASP and AC methods before and after three months of storage at 4, 37 and 42 °C. Results are given as means ± SE.

**Figure 6 toxins-14-00483-f006:**
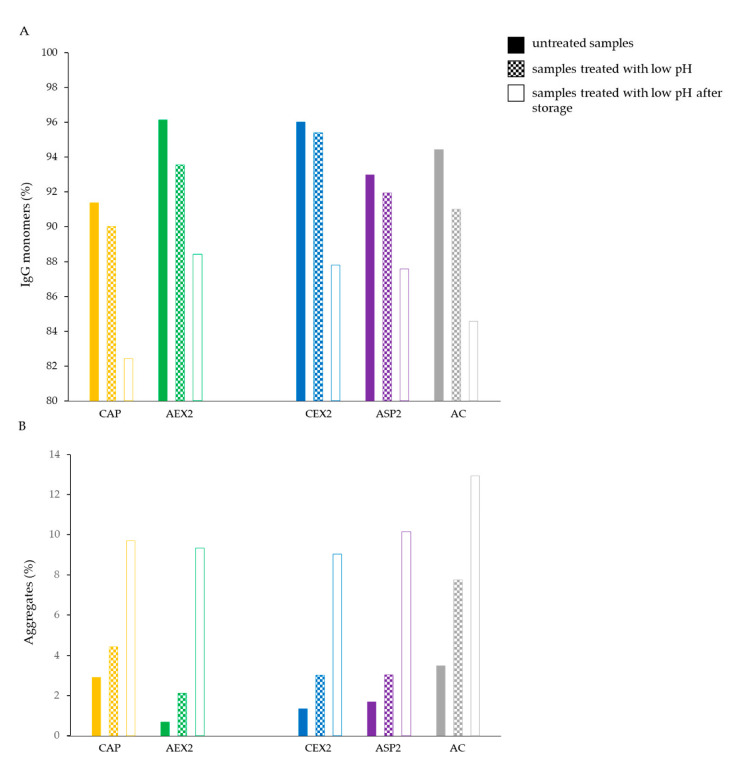
The effect of transient exposure (for 2 h) to low pH conditions (pH 2.0) of IgG samples (*c* = 1 mg mL^−1^) isolated by CAP, AEX, CEX, ASP and AC protocols on IgG monomer (**A**) and aggregate (**B**) content before and after three months of storage at 37 °C.

**Figure 7 toxins-14-00483-f007:**
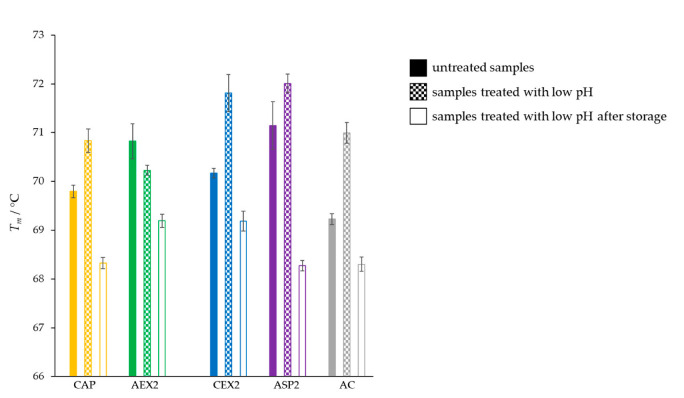
Changes in melting temperatures (*T_m_*) of highly pure IgG preparations (*m* = 25 μg) extracted by CAP, AEX, CEX, ASP and AC protocols (one sample of each type) due to transient exposure (for 2 h) to low pH (2.0) followed by three months of storage at 37 °C. Results are given as the mean of at least 5 measurements ± SE.

**Figure 8 toxins-14-00483-f008:**
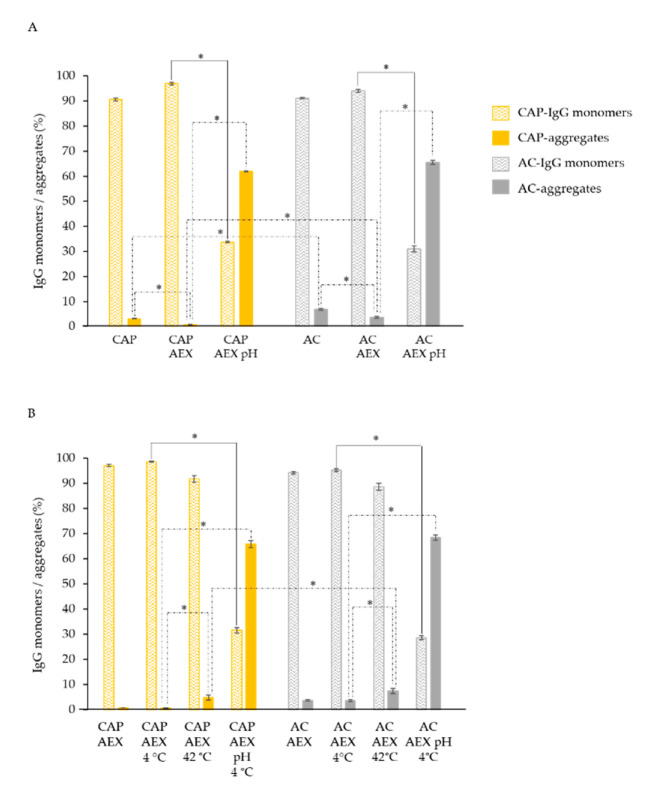
IgG monomer and aggregate content changes due to the exposure to unfavorable conditions before (**A**) and after three months of storage at 4 or 42 °C (**B**). The initial quantity of aggregates was additionally reduced in samples from CAP and AC processing (CAP (*n* = 10) and AC (*n* = 10)) by employing an additional AEX step (CAP AEX (*n* = 10) and AC AEX (*n* = 9)). CAP AEX pH (*n* = 10) and AC AEX pH (*n* = 9) preparations were transiently (for 2 h) exposed to acidic conditions (pH 2.0), after which their pH was returned to the initial value (pH 6.0) following storage at 4 °C for three months. Results are given as means ± SE. * *p* < 0.05.

**Figure 9 toxins-14-00483-f009:**
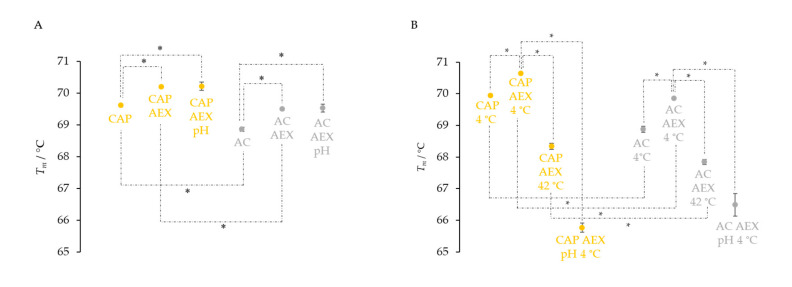
Melting temperature (*T_m_*) changes of pure IgG preparations (*m* = 25 μg) due to exposure to unfavorable conditions before (**A**) and after three months of storage at 4 or 42 °C (**B**). Samples obtained by CAP and AC protocols (CAP (*n* = 10) and AC (*n* = 10)) were additionally purified by the AEX procedure (CAP AEX (*n* = 10) and AC AEX (*n* = 9)). CAP AEX pH (*n* = 10) and AC AEX pH (*n* = 9) samples were transiently exposed (for 2 h) to acidic conditions (pH 2.0), after which their pH was returned to the initial value (pH 6.0) before they were stored at 4 °C for three months. Results are given as means ± SE. * *p* < 0.05.

**Figure 10 toxins-14-00483-f010:**
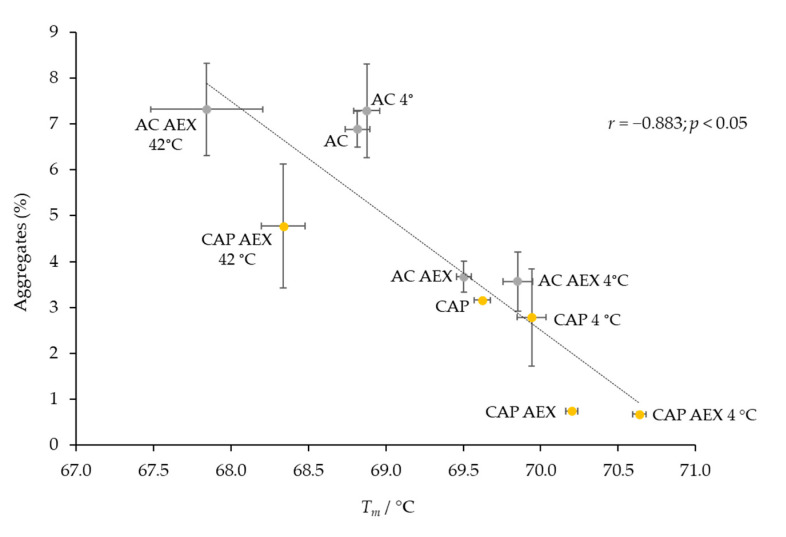
Correlation between *T_m_* values of IgGs and aggregate content in samples obtained by CAP and AC protocols, prior to (CAP (*n* = 10) and AC (*n* = 10)) and after (CAP AEX (*n* = 10) and AC AEX (*n* = 9)) the anion-exchange chromatography step for aggregate removal. Each sample was analyzed at the beginning and end of three months of stability testing at 4 or 42 °C. Individual points are given as means ± SE.

**Table 1 toxins-14-00483-t001:** IgG monomers (%) and aggregates (%) in highly pure samples obtained by caprylic acid precipitation (CAP), anion-exchange chromatography (AEX), cation-exchange chromatography (CEX), ammonium sulfate precipitation (ASP) and protein A affinity chromatography (AC) (one sample of each type) before and after one month of storage at 37 °C, as determined by size-exclusion high-performance liquid chromatography (SEC-HPLC) analysis.

	Before Storage		After Storage	
	IgG Monomers (%)	Aggregates (%)	IgG Monomers (%)	Aggregates (%)
CAP	94.20	1.76	91.50	2.31
AEX2	98.03	0.3	98.31	0.00
CEX2	97.78	1.50	96.06	1.54
ASP2	97.62	1.09	96.17	1.56
AC	89.24	8.82	90.40	6.88

## Data Availability

Original data and Excel files are available on request.

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
