# Peer review of "Roughness of Production Conditions: Does It Really Affect Stability of IgG-Based Antivenoms?"

_toxins, 2022, doi:10.3390/toxins14070483_

Round 1
Reviewer 1 Report
The authors have addressed all previous comments. However, the research design and the significance of the results are still questionable. A more specific hypothesis, methodology and interpretation would be required to give this study novelty value.
Reviewer 2 Report
The authors have improved the manuscript by making significant changes to the wording throughout. There is no document attached that specifically addresses changes raised by reviewer(s), so i can only comment upon the marked up changes presented.
Accelerated stability data such as this is generally not accepted biological product such as antisera, although it is of some interest. Real time stability data is more relevant and is the regulatory accepted stability data. Hence I must interpret very carefully and conclusions made. Heat affects these products markedly, but if stored at recommended conditions of 2-8C is this of any real-world relevance?
This study is very small scale and did not use any charged particle depth filtration of the products, which is extremely important during manufacture to obtain clear and stable product.
Line 300 "...explaination..." Typo.
Round 2
Reviewer 1 Report
While I am still unconvinced regarding the overall merit of the article, I will recommend the article for publication mainly for the following reasoning given by the authors:
"Since their shortage is a well addressed issue, every information which might possibly lead to the even slightest improvement of the production and the final product’s quality is welcomed."
This manuscript is a resubmission of an earlier submission. The following is a list of the peer review reports and author responses from that submission.
Round 1
Reviewer 1 Report
This work is interesting as the subject of adverse reactions from aggregates is often discussed. However this manuscript really needs significant work to better explain the study in a more precise and statistically correct manner. I have provided relatively high level comments as it is difficult to review until this manuscript is significantly reworked.
Title : Harshness of production conditions: does it really count for IgG-based antivenom stability? This does not make sense, needs better wording.
HHP : Why was the plasma heat treated? There was no control of untreated plasma – why? Is the heat treating contributing to aggregation? Is that used during commercial manufacture of the HHP product into antivenom? The need for this step must be questioned.
Abstract: Words that do not seem appropriate are highlighted. The abstract needs significant improvement overall to better explain the study in respect of what problem it is addressing, how the problem was studied and what the discoveries made from the work are. At present it does not do this.
Antivenoms, having pure animal IgGs or their fragments as an active drug, are the only specific therapeutics against envenomation arising from snakebites. Although highly needed, low sustainability of such preparations is causing their constant shortages all over the world due to many reasons. Stability of the product is one of them, contributing not only to sustainability but their safety as well. It has been hypothesized that roughness of conditions to which IgGs are exposed during downstream purification disturbs their conformation to the certain extent, making them accordingly prone to aggregation, particularly after secondary stress exposure. The aim of this research was to investigate the impact of five commonly applied biochemical principles for IgG extraction from hyperimmune plasma on stability properties of pure IgGs. For that purpose, equine IgGs were purified from unique starting material by two mild condition-based operational procedures (anion-exchange chromatography and caprylic acid precipitation) and three harsher ones (ammonium sulphate precipitation, cation-exchange chromatography and protein A affinity chromatography). The stability of refined preparations was studied under non-optimal storage conditions (37 °C, 42 °C, transient lowering of pH) by monitoring the changes in their aggregate content and thermal stability of pure IgGs. Mild purification protocols generated IgG samples with lower aggregate share in comparison to harsher ones. Their tendency for further aggregation was associated with the initial aggregate share. On the contrary, an inverse correlation between the thermal stability of IgG molecules and the aggregate content in refined samples was observed. Since the initial proportion of aggregates in the samples is influenced by the operating conditions, each of them also indirectly affected the stability of final preparations suggesting that mild condition-based refinement protocols indeed generated more stable IgGs.
The report lacks sufficient statistical rigour to justify the conclusions. Until this is corrected it is not possible to make any conclusions with confidence.
Major English language grammatical issues throughout. I would recommend a English as first language speaker review this paper before submitting again.
There is discussion interspersed amongst the results.
Why 37 and 42 degrees Celsius? These are very close together.
Significance of melting temperatures for impure products, is this method applicable for mixed protein solutions? Are these measurements really different?
Reference 12 paper title is incomplete.
How does this research link to antivenom clinical adverse reactions?
Overall this paper needs a lot of work to get it to a standard acceptable for publication.
Reviewer 2 Report
Dear authors,
In general, the grammar and the word choices must be improved throughout the whole manuscript. Here are just a few examples:
Lines 28-31: The sentence is too long and thus hard to follow. Please split it into two shorter sentences.
Line 33-35: Improve the grammar -> One of them is the stability of their active components, a feature that significantly affects the quality as well as safety of these therapeutics.
Line 63-66: The stability of the active drug (please choose a more accurate word, e.g. therapeutic content) in the samples prepared from the unique pool of Vaa-specific hyperimmune horse plasma (HHP) by two mild (CAP, AEX) and three more aggressive (please choose a more relevant word, e.g. harsh) purification protocols (CEX, ASP, AC) formulated in the same buffer was simulntaneously investigated through their exposure to various unfavorable storage conditions followed by monitoring the changes of monomer content and aggregate share, as well as shifts in the melting temperatures (Tm) of IgGs.
The sentence is too long and very hard to follow. Please split it into shorter sentences.
In addition, the Materials and Methods section is coming after the Results, so the Vaa abbreviation needs to be introduced as this is the first time it appears in the text.
Line 74-76: Accordingly, the same sample had the Tm value lover (lowered) by 1.5 °C in comparison to the others whose Tm values were around 70 °C (Figure 1).
Still, the sentence needs to be edited as it's vague in its current form.
Line 351: CH3COONa buffer.
This is the only place where the chemical formula has been used instead of the name of the chemical. Any particular reason?
In materials and methods, I assume, although it's not clearly stated, that the IgG-containing samples were in solution form during the storage time at different temperatures. In that case, how did you deal with the evaporation at 42C for 3 months? Please specify.
Reviewer 3 Report
Major concerns:
· The results of the experiment demonstrate that IgG stability and aggregation is influenced by high salt content, higher temperatures and low pH. Considering that IgG is a protein, these findings are to be expected. While the analysis gave numerical values for these influences, the small number of samples does not provide adequate reliability.
· Results – One-month storage at 37 °C. “Mean from n measurements ± SE”, with n=2 Standard error for 2 measurements is irrelevant both mathematically and statistically. If the intended use of SE was to show the precision of measurements, a more acceptable representation would be to give both individual measured values and their mean. Either way, the sample size is not significant enough for conclusions. As a further question for this table: If before storage there were two samples, why is there a single sample after storage?
· Methods – IgG purification procedures: By using AEX/CAP, CEX/CAP, ASP/CAP the samples were subjected to two subsequent procedures. In the same time, CAP alone represents one of the studied separation methods. As such, the results from AEX2 could be attributed to both AEX and CAP. Is this overlap taken into consideration when presenting the results? (Similarly for CEX2 and ASP2).
· Results – Stability of IgG samples after reduction of initial aggregates content: As above, there are two subsequent separation methods, CAP/AEX, AC/AEX. Has the effect of AEX been taken into consideration? It cannot be considered null, as it is one of the tested separation methods.
· Methods – Thermal shift assay: Line 368 states, “All measurements were performed in pentaplicates.” Caption for figure 1 says “at least 8 measurements”, figure 4 “at least 6 measurements”. How many measurements have been performed? What do you mean by “at least” – why not perform the same number of measurements from each sample and give an exact number? Furthermore, it is not clear if the measurements have been performed n times from the same sample, or there were n samples analysed.
· Methods – Diafiltration: Has the effect of MES buffer and phosphate buffer on IgG aggregation been studied? Since not all samples have been treated equally, some results might be influenced by these buffers as well.
· Methods – Data analysis: Line 391 states, “Number of measurements for each analysis (n) is given.” However, the number of samples is not (with one exception). Were all analysis performed on a single sample for each condition/parameter? If they were, the results have no statistical significance. Kindly state the number of samples used for all determinations. Also amend the captions for figures where needed, as number of samples is not stated (e.g. Figure 8 - Results are given as mean ± SE. * p <0.05. For how many samples/measurements?)
Minor corrections:
· Abstract, Line 11: “conformation to a certain extent”
· Introduction, Line 34: “a feature that…”
· Introduction, Line 41: “with an impact on stability…”
· Introduction, Line 61: “utmost importance”
· Results, Line 65: “was simultaneously investigated”
· Results, Line 81: Since it is here that it first appears in the text, “SEC” abbreviation should be explained.
· Results, Line 104: “In further research, storage time…” There are many commas missing throughout the text, making phrases hard to understand. Kindly check the manuscript for errors.
· Discussion, Line 276: “Namely, increased temperatures…”
· Discussion, Line 287: “that corresponded to…”
· Discussion, Line 295: “for preliminary prediction of stability”
· Discussion, Line 300: “which further progressed”
· Discussion, Line 303: “obvious enhancement of…”
· Discussion, Line 309-310: Please rephrase/simplify this sentence.
· Materials, Line 351: “CH3COONa” should be “sodium acetate”